# $G^3$: Representation Learning and Generation for Geometric Graphs

## Abstract

A geometric graph is a graph equipped with geometric information (i.e., node coordinates). A notable example is molecular graphs, where the combinatorial bonding is supplement with atomic coordinates that determine the three-dimensional structure. This work proposes a generative model for geometric graphs, capitalizing on the complementary information of structure and geometry to learn the underlying distribution. The proposed model, Geometric Graph Generator ($G^3$), orchestrates graph neural networks and point cloud models in a nontrivial manner under an autoencoding framework. Additionally, we augment this framework with a normalizing flow so that one can effectively sample from the otherwise intractable latent space. $G^3$ can be used in computer-aided drug discovery, where seeking novel and optimal molecular structures is critical. As a representation learning approach, the interaction of the graph structure and the geometric point cloud also improve significantly the performance of downstream tasks, such as molecular property prediction. We conduct a comprehensive set of experiments to demonstrate that $G^3$ learns more accurately the distribution of given molecules and helps identify novel molecules with better properties of interest.

## 1 Introduction

Geometric deep learning refers to the development of deep learning techniques for data from non-Euclidean domains, such as graphs and manifolds (Bronstein et al., 2017). In recent years, graph neural networks (GNN) emerged to be a promising family of architectures that models the relational inductive bias ubiquitous to graph structured data (Battaglia et al., 2018; Zhou et al., 2018). With the rise of deep generative modeling, various generative models have been adapted for graphs and proven to be effective in learning the underlying distribution implicitly defined by a set of given graphs (Goodfellow et al., 2014; Kingma & Welling, 2013; Rezende & Mohamed, 2015; van den Oord et al., 2016; De Cao & Kipf, 2018; You et al., 2018b; Jin et al., 2018; Shi et al., 2020).

In this work, we consider a special type of graphs—*geometric graphs*—which are equipped with geometric information in addition to the combinatorial structure (Pach, 2013). In practice, this additional information corresponds to low-dimensional geometric structures and typically appears as node coordinates in $\mathbb{R}^2$ or $\mathbb{R}^3$. The coordinates find broad uses in molecules (Van Aalten et al., 1996), meshes (Alliez et al., 2005), and graph drawing (Frishman & Tal, 2008). This work aims to develop a generative model that captures both the combinatorial and the geometric characteristics exhibited in a given collection of geometric graphs.

One driving application of geometric graphs is molecule generation. In pharmacology, drug discovery is the process of identifying new candidate medications in the form of molecules. The process is known to be difficult, costly, and time-consuming (Paul et al., 2010), because of the discrete nature of the search space and its vast size (estimated to contain at least $10^{33}$ molecules) (Polishchuk et al., 2013). Recently, deep learning techniques have been developed to represent molecules as continuous vectors and apply continuous optimization (Jin et al., 2018) or reinforcement learning (You et al., 2018a) to conduct an efficient search. In almost all existing work based on the molecular-graph approach, the graph is treated as a combinatorial object and the three-dimensional geometric information serves as node features only. However, the atomic coordinates encode vital energy information of the molecule and they can be instrumental for the inference of structure and state. Therefore, a

proper modeling of the geometric information is paramount for a more accurate representation of the molecule in downstream tasks.

In this work, we develop an autoencoder for geometric graphs, orchestrating a GNN and a point cloud model in a nontrivial manner. Besides the straightforward use of a GNN (Duvenaud et al., 2015; Kearnes et al., 2016; Gilmer et al., 2017) to encode the combinatorial structure, we treat the nodes with spatial coordinates as low-dimensional point clouds and use a point cloud model to encode the geometry. Different from decoders merely based on graph structures in conventional graph autoencoders, on the other hand, we decode the graph structure by first reconstructing the geometry. Specifically, we use a folding-based technique (Yang et al., 2018; Pang et al., 2021) to map a template of points to the correct geometry, from which the combinatorial structure is inferred. This way, both sources of information are organically fused and processed. Furthermore, to fulfill the generative capability, we augment the autoencoder with a normalizing flow (Kobyzev et al., 2019; Dinh et al., 2016; 2014), so that one is able to effectively sample from the otherwise intractable latent space.

The proposed geometric graph generator, $G^3$, addresses *geometry* that is rarely considered in past graph generative models. A naive alternative to incorporate node coordinates is to treat them as part of features when consumed by a GNN. This approach, however, biases the geometry by the graph structure owing to local neighborhood aggregation—the global geometry may be compromised. For geometric graphs, the structure itself is combinatorial and discrete, while the geometry is continuous. These two sources of information are often complementary and benefit from collaborative processing. Our strategy is to exploit both graph techniques and point cloud techniques to maximally retain information from the two sources.

To evaluate the effectiveness of $G^3$, we focus on molecule applications and use popular datasets (QM9 and ChEMBL) that contain atomic coordinates. We demonstrate that $G^3$ outperforms a number of representative graph generative models in generating novel and valid molecules and it also significantly outperforms either a GNN or a point model alone for property prediction. Appealingly, $G^3$ decodes molecules substantially faster than do many popular models. Furthermore, with the use of Bayesian optimization in concert with a trained model, we can identify better molecules under metrics of interest than do existing methods.

## 2 RELATED WORK

Generative models refer to a class of machine learning models that can learn the underlying distribution, implicitly defined by a given set of data, and to sample from it. Noteworthy generative models in the deep learning era include variational autoencoders (VAE) (Kingma & Welling, 2013), generative adversarial networks (GAN) (Goodfellow et al., 2014) and normalizing flows (NF) (Kobyzev et al., 2019; Dinh et al., 2016; 2014). Among them, NF bears a unique advantage in estimating the density (likelihood). For training, NF directly optimizes the likelihood, whereas VAE maximizes a lower bound of it (called the ELBO Kingma & Welling (2013)) and GAN minimizes the discrepancy between the input and the transformed noise distribution.

Under these generative models, a graph can be generated in a sequential or one-shot manner: the former samples nodes and edges incrementally while the latter samples a graph in its entirety. Combining GAN and reinforcement learning, MolGAN (De Cao & Kipf, 2018) and GCPN (You et al., 2018a) generate molecules in a one-shot and sequential fashion, respectively. MolecularRNN (Popova et al., 2019) is an autoregressive model based on recurrent neural networks; it performs validity checks during the sequential generation process and rejects invalid structures. JT-VAE (Jin et al., 2018) is a VAE-based tree model that constructs the molecular graph sequentially from sub-components. Popova et al. (2019) point out that JT-VAE may suffer ambiguity in the conversion of the tree structure to a molecular graph, affecting property optimization. GraphNVP (Madhawa et al., 2019) and GraphAF (Shi et al., 2020) are recently proposed NF-based models that parameterize the mapping from a latent vector to a graph by using a flow. The former employs one-shot sampling while the latter is sequential.

Molecules admit representations other than graphs. An alternative is the simplified molecular-input line-entry system (SMILES) (Weininger, 1988), a string notation that universally describes molecular structures. Grammar VAE (GVAE) (Kusner et al., 2017) and syntax-directed VAE (SD-VAE) (Dai et al., 2018) are VAE-based models that process SMILES strings and apply grammatical rules to

sequentially reconstruct the strings. Another family of models, which are motivated by quantum chemistry principles, models molecules as point clouds and learns a physically robust representation to predict, e.g., molecular energies and equilibrium conformations. Schütt et al. (2018) propose SchNet, a network with continuous-filter convolution layers that model interactions between atoms and learn a representation that is invariant to translation and rotation. Using the same filters, Gebauer et al. (2019) devise Generative-SchNet (G-SchNet), which accurately models the target molecule distribution and generates novel 3D conformations that are relatively stable.

In addition to the graph structure, our work treats nodes as a point cloud, which is a collection of points in $\mathbb{R}^3$. Point clouds are a prominent subject in computer vision and computer graphics; they are obtained through, e.g., LIDAR scanning of object surfaces (Yang et al., 2018). Deep learning with point clouds is faced with challenges in defining the convolution operator (Bruna et al., 2013; Bronstein et al., 2017; Schonsheck et al., 2018; Jin et al., 2019). Volumetric CNN (Wu et al., 2015; Qi et al., 2016; Maturana & Scherer, 2015) applies convolution filters on voxels obtained from discretization of the three-dimensional space. Multiview CNN (Su et al., 2015; Qi et al., 2016) reduces point clouds to a collection of 2D images and applies gridded convolutions on them. PointNet (Qi et al., 2017a;b) uses point-wise MLPs to featurize individual points, followed by a symmetric pooling that generates a permutation-invariant global description.

## 3 GEOMETRIC GRAPH GENERATOR

In this section, we present $G^3$. The model is an autoencoder that contains interacting modules to process and reconstruct the geometry (i.e., node coordinates), the structure (i.e., edges and edge types), and additional features (e.g., node features). Such a sophisticated autoencoder allows forming a more accurate representation of the geometric graph and learning better the input distribution.

A proper processing of the geometric and combinatorial information is crucial to the success of $G^3$ and downstream tasks. For example, an authentic representation of the node coordinates serves as the basis for decoding both the node and the edge types: the training hinges on an accurate registration between the reconstructed point cloud and the ground truth. The straightforward adaptation of GNNs for encoding the geometric information (as node features) results in poor coordinate reconstruction and worse generation quality overall. Moreover, one can naively reconstruct coordinates and node features with a single decoder. However, mixing two modes of information together introduces unnecessary bias through, e.g., the difference in scales. Hence, separate but dependent decoders for features and geometry render a more accurate reconstruction.

### 3.1 NOTATION

Let $n$ be the number of nodes, $d_f$ be the number of node features, $d_e$ be the number of edge types, and $d_c$ be the dimension of the node coordinates (typically two or three). We denote a geometric graph as $G = (A, X)$, where $A \in \{0, 1\}^{n \times n \times d_e}$ is the adjacency tensor and $X = [C, F] \in \mathbb{R}^{n \times (d_c + d_f)}$ is the node matrix, where each note vector $x_i = [c_i, f_i]$ is a concatenation of coordinates $c_i \in \mathbb{R}^{1 \times d_c}$ and features $f_i \in \mathbb{R}^{1 \times d_f}$. For molecules, the node features are one-hot encoding of the atom type, and the edge types correspond to bond types.

### 3.2 ENCODER

The purpose of the encoder is to use a latent vector to represent both the geometry and the structure of $G$. It contains two modules: a point encoder that maps the point cloud $C$ to a geometry descriptor $z_p$; and a graph encoder that takes $A$ and $F$ as input and maps them to a structure descriptor $z_g$. Both descriptors are concatenated to form the latent representation $z = [z_p, z_g] \in \mathbb{R}^{d_z}$.

A usual graph encoder processes the structure information $A$ and node features $F$ by using a GNN. When the coordinate information $C$ is available, one could have treated $C$ as part of the node features and feed them into the GNN, indiscriminately of $F$. However, the geometry information of a graph is often complementary to the structure. Hence, a drawback of such an approach is that one heavily relies on the local neighborhoods from graph structures to parameterize the geometry, resulting in a strong bias toward the structure. A notable feature of our design is that we separate the geometry

from the structure when encoding a geometric graph; we will show an ablation study supporting this design subsequently.

This idea offers a means to process all node information, including features $F$ and coordinates $C$, independently of the combinatorial structure, as an additional parameterization over the pure use of a GNN. In what follows, we describe the two encoder modules and defer details of the specific network architecture in the appendix.

**Point Encoder.** In this module, we treat each row of $C$ as a point and use a point cloud architecture to obtain the geometry descriptor $z_p$. First, a multilayer perceptron (MLP) is applied point-wise to the point cloud to amplify the feature dimensions. The point-wise features are then passed through a series of multi-headed attention blocks (Vaswani et al., 2017) to further featurize the points. These attention-based blocks are more effective than simple point-wise MLPs used in PointNet (Qi et al., 2017a) as they are able to leverage global information, i.e. features on all other points, to produce rich and expressive point-wise features. At last, we use a global max-pooling over the points followed by an additional MLP to yield the geometry descriptor $z_p$.

**Graph Encoder.** This module takes the graph adjacency tensor and the node feature matrix as inputs and outputs a structure descriptor $z_g$. Virtually all GNNs serve the purpose; some are even specialized to molecules (e.g., Duvenaud et al. (2015); Kearnes et al. (2016); Gilmer et al. (2017)). Our empirical investigation finds that GINE, Graph Isomorphism Network with Edge features (Xu et al., 2018), obtains a good and stable performance for our purpose.

### 3.3 DECODER

Decoding a graph is substantially harder than encoding one. A common idea is to employ a sequence model and decode nodes and edges one by one Popova et al. (2019). Training such a model is increasingly more challenging for long sequences (i.e., large graphs). On the other hand, approaches decoding the adjacency tensor also face scalability challenges owing to the $O(n^2)$ degrees of freedom Samanta et al. (2020). In this work, we exploit the unique opportunity offered by a geometric graph to decode first the geometry, then the point-wise features, and lastly the structure.

Specifically, we propose a three-step decoding process that decodes a graph $G' = (A', X')$ from $z$, where $X' = [C', F']$. The first step is to decode $C'$, the coordinates for the point cloud, which involves using the latent representation $z$ to morph a template of points to the correct geometry. By isolating the decoding of coordinates, we can leverage specialized architectures to achieve accurate reconstructions of the geometry, which is crucial for the subsequent decoding of $F'$ and $A'$. Secondly, the reconstructed geometry $C'$ is recycled along with the latent code $z$ to predict the point-wise features $F'$. Finally, $X' = [C', F']$ is used to infer the graph structure $A'$ through a link predictor. This procedure sidesteps the scalability challenge of both the sequential generation approach and the adjacency decoding approach.

In what follows, we describe each component of decoder modules and defer details of the specific network architecture in the appendix.

**Coordinate Decoder** The idea of decoding a point cloud from a latent vector $z$ is to successively fold a template of points until it forms the desired geometry. Our coordinate decoder is inspired by TearingNet (Pang et al., 2021), which is a refinement over FoldingNet (Yang et al., 2018; Tao, 2020), a popular archetypal architecture for point cloud reconstruction. Note that the template design is application-dependent; an effective template is guided by the geometry to be modeled. For example, a spherical template with roughly equidistant points is more suitable for the three-dimensional structure of molecules. This design is different from the original TearingNet architecture that uses a regular 2D grid to model surfaces.

**Feature Decoder** To decode the point-wise features $F'$, we apply the folding operations with the decoded coordinates $C'$ from the aforementioned coordinate decoder as the starting template. The latent code $z$ is replicated and concatenated to $C'$, which is then forwarded through point-wise MLPs to complete one set of folding. We repeat two sets of folding to arrive at the decoded features. The architecture of the feature decoder is illustrated in Figure 1.

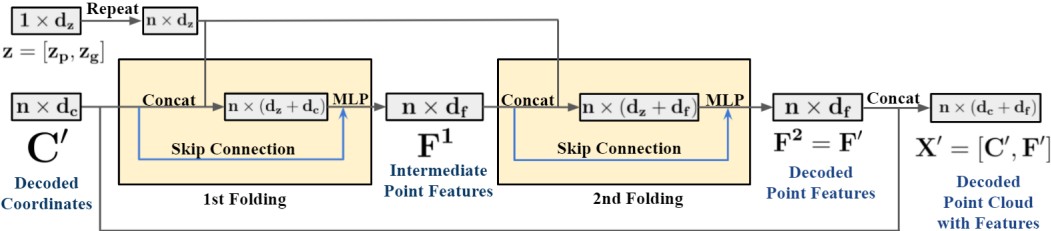

**Figure 1:** The $G^3$ feature Decoder.

**Link Predictor (Graph Decoder)** The link predictor determines the existence of an edge (and its type if so) for every pair of decoded nodes. A typical link predictor takes the inner product between two nodes to indicate the strength of connection (Trouillon et al., 2016). However, such a link predictor is oblivious to the geometry because a large inner product in 3D does not necessarily indicate closeness. Rather, the displacement of two points, their respective neighborhoods, and their neighbors' features may all play a role in determining the existence and type of an edge. For example, in molecules, the bond between a pair of atoms depends on their distance as well as the neighboring atoms. Therefore, we augment the features of a decoded node $x_i$ from $[c_i, f_i]$ to $\hat{x}_i$ as

$$\hat{x}_i = [|c_i - c_{n_1}|, ..., |c_i - c_{n_{k-1}}|, f_i, f_{n_1}, ..., f_{n_{k-1}}],$$

where $n_1, \ldots, n_{k-1}$ are the node indices of the sorted $k$-nearest neighbors of $x_i$; $f_i \in \mathbb{R}^{d_f}$ and $c_i \in \mathbb{R}^{d_c}$ are decoded features and coordinates, respectively; and the absolute sign indicates element-wise absolute values. We then use these augmented features as well as the coordinate-wise displacement $|c_i - c_j|$ to define the link predictor. Formally,

$$l(i, j) = \text{MLP}([\hat{x}_i, \hat{x}_j, |c_i - c_j|]) \in \mathbb{R}^{d_e},$$
$$\text{LinkPredictor}(i, j) = l(i, j) + l(j, i) \in \mathbb{R}^{d_e}.$$

We highlight that our link predictor works remarkably well as a standalone module for molecules, which can be characterized as a procedure for inferring inter-atomic bonds based on molecular conformation and atom types. To the best of our knowledge, there are no reliable public software available for this task, especially for large molecules with noise in coordinates or atom types. Therefore, our proposed link predictor has its own value other than being a part of $G^3$. We refer appendix G for illustrating the outstanding performance of the standalone link predictor.

### 3.4 LATENT SPACE SAMPLING: NORMALIZING FLOW

To reliably sample from $G^3$'s latent space, we make use of a normalizing flow model (Kobyzev et al., 2019), which consists of a sequence of invertible transformations that map between an easily sampled base distribution (e.g., standard normal) and an unknown distribution (e.g., the latent distribution of our autoencoder). The densities of the two distributions are related by the Jacobian of the transformations, so that log-likelihood of the unknown distribution can be computed and optimized for training. In practice, we find that RealNVP (Dinh et al., 2016) to be a simple yet effective model for our needs. After the training is finished, samples from the base distribution are forwarded through the normalizing flow and subsequently decoded into the generated molecules.

### 3.5 OVERALL MODEL AND TRAINING LOSS

The overall architecture of $G^3$ is outlined in Figure 2, which summarizes all the components described in the preceding subsections. To train this model, we use a loss consisting of multiple terms with a tunable weighting scheme: $L = \lambda_1 \cdot L_C + \lambda_2 \cdot L_F + \lambda_3 \cdot L_E + \lambda_4 \cdot L_{NF} + \lambda_5 \cdot L_R$.

**Node coordinate loss $L_C$.** In this loss, we use only the coordinate part $C$ contained in the node matrix $X$. In our experiments, we use the Sliced Wasserstein-2 distance (SW2) as our loss for robust point cloud reconstructions (Rabin et al., 2011). Note that in the calculation of $L_C$, every point $c \in C$ is identified with the nearest match $c' \in C'$ and vice versa. This correspondence allows us to feasibly define the following losses $L_F$ and $L_E$ with differentiability. We remark that another commonly used loss for point-cloud matching, Chamfer distance (Yang et al., 2018), can also be used here.

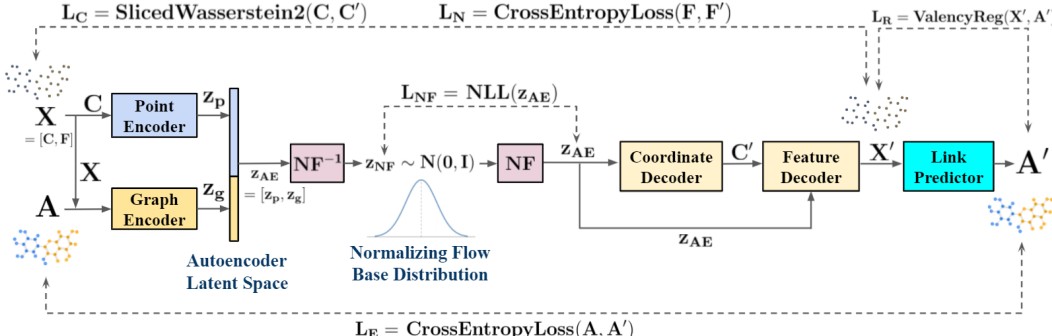

**Figure 2:** Overall architecture of $G^3$.

**Node feature loss $L_F$.** Since node correspondence is established, the difference between $F$ and $F'$ can be meaningfully quantified. When the rows are one-hot vectors, a cross-entropy loss is suitable; otherwise, mean absolute error (MAE) or other losses may be appropriate.

**Edge loss $L_E$.** The edge loss quantifies whether edge types are correctly inferred. Hence, a cross-entropy loss between $A$ and $A'$ is natural, where node correspondence is needed.

**Normalizing flow loss $L_{NF}$.** The normalizing flow model is equipped with a standard negative log-likelihood loss (Dinh et al., 2016).

**Additional loss $L_R$.** This term enforces structural and other chemical constraints innate to the specific application. Some constraints may be easily violated even if the decoding of nodes and edges is reasonably accurate (as long as error exists, however small). Hence, a regularization that penalizes the violation of the constraints helps improve semantic validity. For example, for molecules, one incorrect decoding of a carbon atom to oxygen is enough to create an invalid molecule, because the valence of oxygen is much smaller. In this case, we follow the work by Ma et al. (2018) and include a soft valence regularization term $L_R$ to encourage higher validity of the generated molecules.

## 4 EXPERIMENTS

### 4.1 DATASETS AND BASELINE MODELS

We perform experiments on two benchmark datasets that come with atomic coordinate information: QM9 (Ruddigkeit et al., 2012; Ramakrishnan et al., 2014) and ChEMBL (Mendez et al., 2019; Davies et al., 2015). The QM9 dataset contains 134k molecules with 4 atom types and up to 9 heavy atoms per molecule. The ChEMBL dataset contains 1.5 million molecules, and the largest one has hundreds of atoms. For feasible experimentation, we take a subset of it that contains 250k molecules with 19 atom types and up to 39 heavy atoms per molecule. We name this new dataset ChEMBL250k. Details of dataset proprocessing are provided in the appendix.

We compare $G^3$ with several representative architectures that feature a variety of generative approaches. SDVAE (Dai et al., 2018) is a syntax-directed VAE model that processes SMILES representation of molecules and generates syntactically correct ones. MolGAN (De Cao & Kipf, 2018) combines GAN and reinforcement learning to generate molecules with optimized properties. JT-VAE (Jin et al., 2018) is a VAE-based model that assembles chemical substructures, boasting high validity of the generated molecules. GraphNVP (Madhawa et al., 2019) is a flow-based model with a one-shot generation procedure. GraphAF (Shi et al., 2020) is another flow-based autoregressive model that advertises high flexibility in density modeling. G-SchNet (Gebauer et al., 2019) is a quantum-chemistry-inspired model that learns accurate conformations and sequentially generates molecules close to the trained distribution.

### 4.2 GENERATION QUALITY

**Metrics and processing.** To evaluate the generative capability of different models, we measure the validity, uniqueness, and novelty (V, U, N) on their generated molecules. Validity is the percentage of

molecules that pass RDKit's sanitization check. Uniqueness is the portion of valid molecules that are distinct. Novelty is the portion of valid molecules that do not appear in the training set.

In addition, we include metrics computed between the test and the generated molecules via MOSES (Polykovskiy et al., 2020): Frechet ChemNet Distance (FCD), which measures the distance between the two sets according to ChemNet's penultimate layer's activation; Fragment similarity (FRAG) measures the similarity in fragments present in both sets of molecules; Internal Diversity (IntDiv2) assesses the diversity within the generated molecules; LogP, QED, and Weight are the 1D Wasserstein distances between the two sets on different chemical properties: LogP measures the lipophilicity of a compound (Bhal, 2007); QED is the quantitative estimate of *druglike-ness* (Bickerton et al., 2012); Weight is the sum of atomic weights in a molecule. All scores are computed on 10k generated molecules.

In our experiments, $G^3$ sometimes generates disconnected molecules, for which we treat the largest connected component as the final generated molecule (breaking ties arbitrarily) for evaluation. Additionally, we observe the normalizing flow cannot capture the autoencoder's latent distribution perfectly when trained on ChEMBL250k. Therefore, we interpolate between samples from the normalizing flow and the nearest encoded training data before decoding.

| MODEL | V↑ | U↑ | N↑ | FCD↓ | FRAG↑ | INTDIV2↑ | LOGP↓ | QED↓ | WEIGHT↓ |
|---|---|---|---|---|---|---|---|---|---|
| SDVAE | 33.0 | 73.6 | 93.4 | 8.52 | 0.57 | 0.870 | 2.50 | 0.062 | 206.82 |
| MOLGAN | 74.5 | 23.1 | 52.5 | 3.69 | 0.66 | 0.869 | 0.63 | 0.028 | 2.17 |
| JT-VAE | 100 | 70.2 | 43.3 | 0.94 | 0.94 | 0.894 | 0.14 | 0.016 | 19.50 |
| GRAPHNVP | 86.6 | 83.5 | 56.5 | 3.83 | 0.86 | 0.886 | 1.06 | 0.027 | 1.04 |
| GRAPHAF | 100 | 69.8 | 86.1 | 7.86 | 0.22 | 0.826 | 0.74 | 0.057 | 62.83 |
| G-SCHNET | 79.2 | 94.1 | 63.9 | 1.99 | 0.90 | 0.892 | 0.38 | 0.007 | 5.06 |
| $G^3$ | 92.7 | 97.0 | 72.2 | 1.23 | 0.94 | 0.900 | 0.25 | 0.006 | 1.60 |

**Table 1:** Molecular scores on the QM9 dataset.

| MODEL | V↑ | U↑ | N↑ | FCD↓ | FRAG↑ | INTDIV2↑ | LOGP↓ | QED↓ | WEIGHT↓ |
|---|---|---|---|---|---|---|---|---|---|
| SDVAE | 32.8 | 81.9 | 94.0 | 28.09 | 0.47 | 0.889 | 5.25 | 0.213 | 331.57 |
| MOLGAN | 100 | 0.02 | 100 | 48.52 | 0.01 | 0.270 | 1.04 | 0.403 | 284.11 |
| JT-VAE | 100 | 97.8 | 99.9 | 7.39 | 0.86 | 0.854 | 0.84 | 0.189 | 72.45 |
| GRAPHNVP | 31.6 | 95.8 | 99.8 | 39.62 | 0.13 | 0.771 | 1.70 | 0.271 | 69.94 |
| GRAPHAF | 100 | 72.0 | 91.9 | 26.95 | 0.46 | 0.895 | 1.39 | 0.117 | 182.12 |
| $G^3$ | 78.2 | 99.4 | 94.1 | 7.28 | 0.95 | 0.893 | 0.59 | 0.067 | 64.10 |

**Table 2:** Molecular scores on the ChEMBL250k dataset.

**Results and discussions.** Results reported in Tables 1 and 2 invite a few observations. Overall, we see that $G^3$ has strong performances across all metrics in both QM9 and ChEMBL250k. Notably, $G^3$'s low FCD score indicates accurate modeling of the training distribution, while a high novelty rate means $G^3$ has not simply memorized the training set. This is important because a model can always reproduce the training set to get perfect scores on e.g. FCD. In contrast, although JT-VAE has better FCD and LogP scores on QM9, its low novelty is characteristic of overfitting. In addition, $G^3$'s validity scores are consistently the highest among the models with no hard valency checks (the others have 100% validity), and $G^3$'s high uniqueness and molecular diversity suggest it is able to avoid mode collapse, which is evident in MolGAN. Furthermore, $G^3$'s robust performance in LogP, QED, and Weight implies that $G^3$'s generated molecules are not only faithful to the training set in terms of (sub)structure, but also with respect to various chemical properties.

**Efficiency.** $G^3$'s performance on large molecular datasets is especially impressive considering its one-shot generation approach and is far superior to other models in the same camp (GraphNVP, MolGAN). Comparing to the best sequential models, e.g. JT-VAE, $G^3$ can generate molecules as good in quality in a much faster fashion, thanks to its one-shot nature. When benchmarked on a machine with i7-8700K CPU, GeForce 1080 RTX GPU, and 16GB RAM, GraphAF and JT-VAE took 4804.36s and 2642.97s to generate 10k molecules when trained on ChEMBL250k, respectively. In contrast, $G^3$ took 47.79s, which is approximately 55 times faster than JT-VAE and 100 times faster

than GraphAF. As discussed, the inefficiency of sequential generation models is mainly due to the challenge of parallelization.

**Latent space interpolation.** Lastly, we present examples of $G^3$'s generated molecules as well as their local latent space in Figure 3. The surrounding molecules are decoded along two random orthogonal axes starting from the center molecule highlighted by the blue box. The transition between molecules are gradual and chemically sound, which illustrates $G^3$'s great capability to smoothly parameterize discrete molecules in its latent space. More qualitative examples can be found in the appendix.

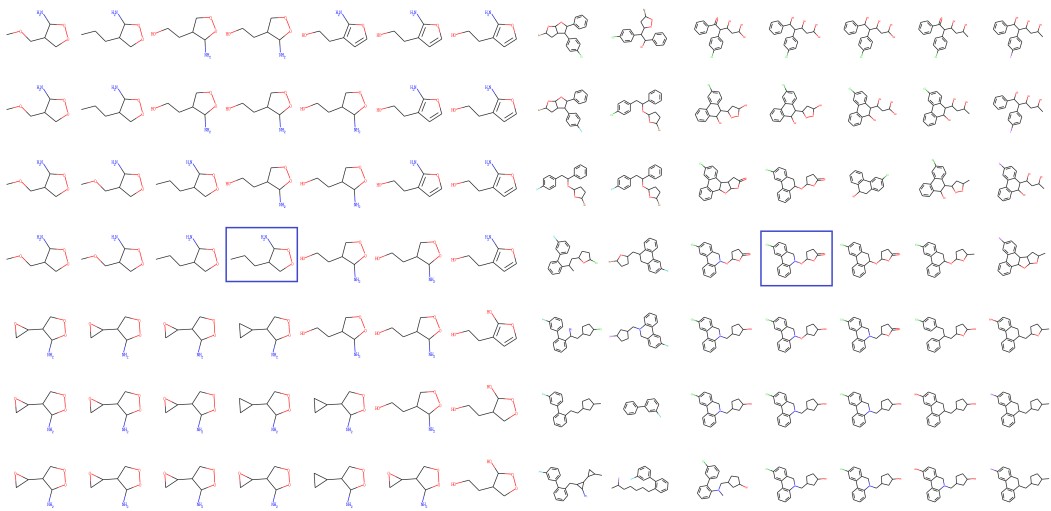

**Figure 3:** Latent space interpolation for QM9 (left) and ChEMBL250k (right).

## 4.3 Ablation Study and Property Prediction

To process the complementary geometry and structure information, we use a pair of point encoder and graph encoder for $G^3$. A natural question asks if two encoders are really necessary. Here, we conduct an ablation study and compare $G^3$ with two simpler models. The first one uses only the point encoder and removes the graph encoder. Because of a lack of graph structure encoding, the edge decoding quality entirely depends on node information. The second simplified model is to remove the point encoder but retain the graph encoder. Since the graph encoder takes coordinates as part of the input, the geometry information is still encoded, albeit through a GNN module that was not designed to handle geometry in the first place.

| Encoder | V↑ | U↑ | N↑ | FCD↓ | Frag↑ | IntDiv2↑ | LogP↓ | QED↓ | Weight↓ |
|---------|------|------|------|------|-------|----------|-------|-------|---------|
| Point | 93.0 | 96.5 | 74.6 | 1.67 | 0.90 | 0.897 | 0.22 | 0.006 | 2.84 |
| Graph | 92.6 | 95.2 | 72.4 | 1.57 | 0.92 | 0.897 | 0.34 | 0.008 | 2.74 |
| Both | 92.7 | 97.0 | 72.2 | 1.23 | 0.94 | 0.900 | 0.25 | 0.006 | 1.60 |

**Table 3:** Ablation molecular scores on the QM9 dataset.

We train the models on QM9 until convergence using the same hyperparameters and latent dimensions. From Table 3, $G^3$'s encoder achieves the best results in two-thirds of the metrics, and the remaining third is split between the other approaches. We further highlight the combined encoder's convincing lead in FCD, which is the best single indicator of the quality of generated molecules in the absence of overfitting. As a result, we find $G^3$'s combined approach to be clearly favored.

To further indicate that the combination of geometry and graph information used in $G^3$ learn a meaningful latent representation for the molecules, we show downstream tasks for predicting chemical properties. In the subsequent experiment, we show that a $G^3$ encoder pretrained in a unsupervised manner can be used to predict chemical properties of QM9 molecules with an additional MLP to fit

| MODEL | H-L $\downarrow$ | INT. ENERGY $\downarrow$ | FREE ENERGY $\downarrow$ | HEAT CAPACITY $\downarrow$ |
|---|---|---|---|---|
| PRETRAINED GRAPH ENCODER | 0.099 | 0.040 | 0.040 | 0.050 |
| PRETRAINED $G^3$ ENCODER | 0.091 | 0.026 | 0.026 | 0.049 |
| GRAPH ENCODER | 0.033 | 0.014 | 0.020 | 0.013 |
| $G^3$ ENCODER | 0.027 | 0.006 | 0.005 | 0.006 |

**Table 4:** Relative MAE on the test set for QM9 property prediction.

desired properties. Alternatively, one can also train $G^3$'s encoder in a supervised manner to predict the desired properties, which yield better results. The pretrained models are first trained on the QM9 dataset in an unsupervised manner for 80 epochs. The weights on the encoders are then frozen, and an additional MLP is trained on the encoded vectors until convergence. The supervised models are trained on the QM9 dataset until convergence.

We predict four properties that come with the QM9 dataset: Homo-Lumo energy (H-L), internal energy, free energy, and heat capacity. In all cases, $G^3$'s graph-point encoder significantly outperforms the standalone graph encoder used in section 3.2. This further substantiates $G^3$'s ability to leverage the geometric and combinatorial features in a geometric graph more effectively by separating the graph and the point encoder.

## 4.4 PROPERTY OPTIMIZATION

One use of generative models with latent spaces is to search for better molecules under metrics of interest. Following the approach outlined by Kusner et al. (2017), we apply Bayesian optimization (BO) on the $G^3$ model pretrained from Table 1 and 2 to optimize two metrics: penalized logP and QED. Penalized logP is logP less the synthetic accessibility score and the number of long cycles (Bhal, 2007), and QED measures how likely a compound is to be a drug (Bickerton et al., 2012).

| | QM9 | | | | | | CHEMBL250K | | | | | |
|---|---|---|---|---|---|---|---|---|---|---|---|---|
| | PENALIZED LOGP | | | QED | | | PENALIZED LOGP | | | QED | | |
| MODEL | 1ST | 2ND | 3RD | 1ST | 2ND | 3RD | 1ST | 2ND | 3RD | 1ST | 2ND | 3RD |
| SDVAE | 4.96 | 4.85 | 4.12 | 0.528 | 0.515 | 0.509 | 5.04 | 2.09 | 1.95 | 0.843 | 0.809 | 0.806 |
| GRAPHAF | 2.50 | 2.24 | 2.24 | 0.593 | 0.588 | 0.584 | 3.09 | 2.53 | 2.49 | 0.830 | 0.821 | 0.820 |
| JT-VAE | 4.99 | 4.66 | 4.38 | 0.623 | 0.621 | 0.620 | 5.34 | 4.85 | 4.70 | 0.935 | 0.934 | 0.932 |
| $G^3$ | 5.97 | 5.64 | 5.37 | 0.640 | 0.621 | 0.614 | 5.46 | 4.94 | 4.82 | 0.941 | 0.929 | 0.927 |

**Table 5:** Best scores found for QM9 and ChEMBL250k.

The results for QM9 and ChEMBL250k are reported in Table 5. One sees that $G^3$ consistently finds the best molecule under both metrics. We remark that BO works better for low dimensional search spaces (Frazier, 2018). The latent spaces of SDVAE and JT-VAE are of a similar nature to that of $G^3$, but GraphAF must use a latent space of the same dimension as the input space. Hence, BO may not be the best method for GraphAF. Indeed, Shi et al. (2020) employs reinforcement learning to chase for optimal metric values, which gives more promising results on ZINC250k. However, for fairness of comparison, we opt to use BO for all methods. More details are provided in the appendix.

## 5 CONCLUSION

We have presented a generative model $G^3$ for geometric graphs that are equipped with additional geometry information. Geometric graphs appear in many practical applications, a notable one being molecules, whose three-dimensional structure is rarely captured by existing graph generative models. Our model is an autoencoder that employs both GNN architectures and point cloud architectures to handle the complementary structure and geometry information. Moreover, a normalizing flow is inserted into the latent space so that one can effectively sample new graphs. Experiment results show that $G^3$ is able to more accurately learn the distribution of given graphs and better predict molecular properties. It helps discover novel molecules with better properties than do many representative graph generative models.

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

# A    EXPERIMENTAL DETAILS

**Datasets**   We preprocess each molecule by centering it based on the 3D coordinates and aligning its orientation with the principal axes via PCA. The molecules in the datasets have different number of atoms. Hence, we pad each graph with "phantom nodes" to reach the maximum molecule size in the dataset. The phantom nodes are indicated by one of the node features, and we also treat nonexistence of edges as an edge type.

Importantly, the phantom nodes need artificial coordinates. We cannot naively allocate all the phantom nodes to the origin, because overlapping coordinates cause non-unique correspondences in the point-cloud registration, making the learning of atom and edge types impossible. In practice, we use an extra coordinate dimension and assign the phantom coordinates for the $i$-th phantom node as $(0, 0, i)$ or $(0, 0, 0, i)$ if the original point cloud is in 2D or 3D, respectively.

In addition, we make a distinction between each atom type and its aromatic counterpart whenever possible. For example, a carbon atom may be part of an aromatic ring, in which case we mark its atom type as aromatic carbon to distinguish it from an ordinary carbon atom. This facilitates the reconstruction of ring structures in the link predictor, as it can predict aromatic bonds easily by checking if the atom types are aromatic.

Note that another popularly used benchmark, ZINC250k (Sterling & Irwin, 2015), does not contain coordinate information and hence we omit it. However, the scales of ChEMBL250k and ZINC250k are rather close, and their results unsurprisingly share similar trends when cross-checked against the performance of baseline models.

**Models**   For G$^3$, we use the same architecture and identical number of layers and hidden dimensions on both QM9 and ChEMBL250k, without separate fine tuning. Such a practice helps corroborate robustness of the model under different data size and complexity. The latent dimension used for QM9 is $[d_p, d_g] = [10, 30]$, and $[35, 25]$ for ChEMBL250k. The additional graph latent dimensions is required to accurately decode the diverse atom types present in ChEMBL250k, which is more than twice than that in QM9.

| NAME | MODEL | GENERATION | REPRESENT. |
|---|---|---|---|
| SDVAE | VAE | SEQUENTIAL | SMILES |
| MOLGAN | GAN | ONE-SHOT | GRAPH |
| G-SCHNET | AR | SEQUENTIAL | POINT CLOUD |
| JT-VAE | VAE | SEQUENTIAL | GRAPH |
| GRAPHNVP | NF | ONE-SHOT | GRAPH |
| GRAPHAF | NF | SEQUENTIAL | GRAPH |
| G$^3$ | NF | ONE-SHOT | GEOMETRIC GRAPH |

**Table 6:** Summary of baseline models.

We did not check G-SchNet's performance on ChEMBL250k for two reasons. First, G-SchNet excels at reconstructing and generating conformers, which are the 3D structures of molecules that are crucial for various computations in quantum chemistry. The coordinates of ChEMBL250k, however, come from molecular graphs and are all in 2D. Second, G-SchNet's sequential generation process utilizes all previously placed atoms to predict the position of the new atom at each step, which scales poorly with respect to the size of molecules. As many ChEMBL250k molecules have more than 100 atoms after adding Hydrogens, the computational resource required to train a G-SchNet model on ChEMBL250k is likely to be prohibitively high.

All baseline results are obtained by running the implementation provided by the authors, with default hyperparameters. The baseline results generally match the reported ones from the original papers. One discrepancy occurs on GraphAF with QM9, which is caused by the disabling of a filter that ignores small molecules. Turning off the filter amounts to a more fair comparison because small molecules are still sensible results. The V, U, N metrics for MolGAN on QM9 are also somewhat off, which is likely a symptom of GAN's training instability.

**Code** $G^3$ is implemented with the Pytorch framework. All relevant code as well as the trained models will be made available upon publication.

## B  ARCHITECTURE DETAILS

In this section, we specify the architecture used to produce the reported experiment results. Let $\mathbf{MLP}(n, h, o, l)$ denote a (point-wise) multi-layer perceptron that has input dimension $n$, hidden dimension $h$, output dimension $o$, and number of hidden layers $l$. Let $\mathbf{MHA}(n, h)$ be the multi-headed attention block with embedding dimension $n$ and $h$ heads. The implementation of multi-headed attention is borrowed from Paszke et al. (2019).

**Point Encoder** As illustrated in Figure 4, the $G^3$ point encoder is :

$$\mathbf{MLP}(n, 256, 768, 1) \rightarrow \mathbf{MHA}(256, 3) \rightarrow \mathbf{MHA}(256, 3) \rightarrow \mathbf{GlobalMaxPooling}$$
$$\rightarrow \mathbf{MLP}(768, 256, d_p, 1)$$

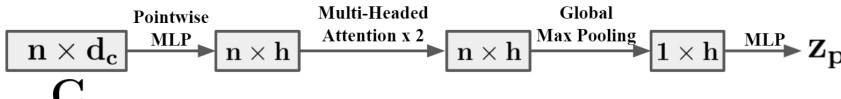

**Figure 4:** The $G^3$ Point Encoder.

**Graph Encoder** Let $\mathbf{GINE}(h, l)$ denote a GINE kernel (Xu et al., 2018) with embedding dimension $h$ and $l$ hidden layers in the MLP. Let $\mathrm{Set2Set}$ be a Set2Set global pooling layer (Vinyals et al., 2015). We first use linear layers to embed the node and edge features to 256 dimensional vectors. The graph encoder is

$$\mathbf{GINE}(256, 3) \rightarrow \mathbf{GINE}(256, 3) \rightarrow \mathbf{GINE}(256, 3) \rightarrow \mathbf{Set2Set} \rightarrow \mathbf{MLP}(512, 256, d_g, 2)$$

A Batchnorm (Ioffe & Szegedy, 2015) layer is applied after every GINE convolution.

**Coordinate Decoder** This module is inspired by (Pang et al., 2021). Let $U$ be the template with the $i$-th row denoded as $u_i$, $z = [z_p, z_g]$ be the latent vector, and $Y'$ be the output with the $i$-th row denoded as $y'_i$. Define the FoldingNet $\mathbf{F}$, where $\mathbf{F}(U, z) = Y'$. It performs the following operation for each point $i$:

$$y_i^1 \leftarrow \mathbf{MLP}(u_i, z)$$
$$y_i^2 \leftarrow \mathbf{MLP}(y_i^1, z)$$
$$y_i' \leftarrow \mathbf{MLP}(y_i^2, z)$$

Define the TearingNet $\mathbf{T}$, where $\mathbf{T}(U, X, z) = Y'$. It performs the following operation for each point $i$:

$$y_i^1 \leftarrow \mathbf{MLP}(u_i, x_i, z)$$
$$y_i' \leftarrow \mathbf{MLP}(u_i, y_i^1, x_i, z)$$

The coordinate decoder does the following operations:

$$C^1 \leftarrow \mathbf{F}(U, z)$$
$$U^1 \leftarrow \mathbf{T}(U, X^1, z) + U$$
$$C' \leftarrow \mathbf{F}(U^1, z)$$

Where $C'$ with the $i$-th row as $c'_i$ are the decoded coordinates. The MLPs all have two hidden layers with hidden dimensions 1024.

**Feature Decoder**    Let $C'$ be the decoded coordinates with the $i$-th row denoded as $c'_i$, and $z = [z_p, z_g]$ be the latent vector. Using the same FoldingNet notation as above, the feature decoder does the following:

$$F' \leftarrow \mathbf{F}(C', z)$$

Where $F'$ with the $i$-th row as $f'_i$ are the decoded features. The MLPs all have two hidden layers with hidden dimensions 1024.

**Link Predictor**    Let $d_x = 2((d_a + d_c) \cdot k) - d_c$. The MLP inside the link predictor is $\mathbf{MLP(d_x, 1024, d_e, 3)}$.

## C    HYPERPARAMETERS

**Experiment on generation quality**    $G^3$: For QM9, the latent vector $z = [z_p, z_g]$ has dimension $10 + 30$ and the loss weighting is $(\lambda_1, \lambda_2, \lambda_3, \lambda_4, \lambda_5) = (10, 1, 1, 10^{-3}, 10)$. For ChEMBL250k, the latent dimension is $35 + 25$ and the loss weighting is $(10, 0.1, 9, 10^{-4}, 5)$.

GraphAF: In this model, there is an option to filter out generated molecules with fewer than $m$ atoms. Since no other models use filtering, we set $m = 0$ for a fair comparison. Because larger molecules are more likely to be novel and unique, the scores reported by the original authors are higher than those we obtain. Moreover, we have tried best efforts to tune hyperparameters for QM9 as the authors did not release their setting.

Other models: We use the default hyperparameters released by the authors for QM9. For ChEMBL250k, we borrow those used for ZINC250k. We find this to be a reasonable approach because ZINC250k and ChEMBL250k have the same number of molecules, and the maximum number of atoms per molecule is also quite close: 38 and 39, respectively. As such, it is expected that the resulting performance on these two datasets is similar, which is indeed the case.

**Experiment on property optimization**    In each iteration of the Bayesian optimization, a Gaussian process is used to fit the explored data, and a batch of new candidates are suggested based on the use of an acquisition function. For all models, we use training molecules as the initial set of points and perform five iterations of Bayesian optimization, wherein the acquisition function is EI (expected improvement). In each iteration, 60 candidates are generated.

The implementation is provided by Kusner et al. (2017) via a custom build of Theano. To conduct Bayesian optimization on the baseline models, we use the default hyperparameters released by the authors whenever available, and follow the practice above otherwise.

## D    LOSS DETAILS

For completeness, in what follows we give the detailed expressions for all the training losses. They are generally straightforward or have been defined in the literature.

### D.1    COORDINATE LOSS

The Wasserstein distance is intimately connected with the study of Optimal Transport. Let $(\Omega, \mathcal{F})$ be a measurable space, and $\mathcal{P}(\Omega)$ be the set of all probability measures on $(\Omega, \mathcal{F})$. Define two probability measures $\mu, \nu \in \mathcal{P}_2(\Omega)$, where $\mathcal{P}_2(\Omega) = \{\mu \in \mathcal{P}(\Omega) : \int_\Omega \|x\|^2 \mu(dx) < \infty\}$. The Wasserstein-2 distance $W_2(\mu, \nu)$ is defined as:

$$W_2^2(\mu, \nu) = \inf_{\gamma \in \mathcal{C}(\mu, \nu)} \int_{\Omega \times \Omega} \|x - y\|^2 \gamma(dx, dy)$$

Where $\mathcal{C}(\mu, \nu)$ is the set of all couplings between $\mu, \nu$ that satisfies: $\gamma(A \times \Omega) = \mu(A), \forall A \in \mathcal{F}$, $\gamma(\Omega \times B) = \mu(B), \forall B \in \mathcal{F}$, and $\mathcal{F}$ is the sigma algebra associated with $\Omega$. It is known that $W_2(\mu, \nu)$ defines a metric on $\mathcal{P}(\Omega)$ (Ambrosio et al., 2008).

In general, $W_2(\mu, \nu)$ is difficult to compute, especially in high dimensions. However, it has a simple form in one-dimension:

$$W_2(\mu, \nu) = \int_0^1 |F_\mu^{-1}(\tau) - F_\nu^{-1}(\tau)|^2 d\tau$$

Where $F_\mu, F_\nu$ denote the cumulative distribution functions associated with $\mu, \nu$, respectively. The simple one-dimensional form of $W_2(\mu, \nu)$ inspires the usage of *sliced Wasserstein distance*, which is the expected value of the projected one-dimensional Wasserstein distance. Define the push-forward operation $*$ as: $f_*\mu(A) = \mu(f^{-1}(A)), \forall A \in \mathcal{F}$, and the projection map as $\Theta(x) = \langle \theta, x \rangle$ for a normalized direction $\theta$. Formally, the sliced Wasserstein-2 distance $SW_2(\mu, \nu)$ in $d$-dimension is:

$$SW_2(\mu, \nu) = \int_{\mathbb{S}^{d-1}} W_2(\Theta_*(\mu), \Theta_*(\nu)) d\theta$$

where $d\theta$ is the uniform measure on $\mathbb{S}^{d-1}$. Bonnotte (2013) has shown that $SW_2(\mu, \nu)$ also defines a metric on $\mathcal{P}(\Omega)$, and it induces the same topology as $W_2(\mu, \nu)$ when $\Omega$ is compact.

In practice, we compute $SW_2(\mu, \nu)$ by approximating the integral via Monte Carlo sampling directions on the sphere. We then compute the projection of the point clouds onto the sampled directions, sort the one-dimensional projections, and average their differences. Through experimentation, we found that 200 is an appropriate number of directions to sample.

## D.2 Node Feature Loss $L_F$ and Edge Loss $L_E$

Let $C \in \mathbb{R}^{n \times d_c}$ be the original point cloud, where each row $c_i := C_{i,:}$ gives the coordinates of a point. Similarly, let $C' \in \mathbb{R}^{n \times d_c}$ be the reconstructed point cloud with $c'_i := C'_{i,:}$. Define the permutation matrix $\sigma$ as:

$$\sigma_{ij} = \begin{cases} 1, & \text{if } j = \arg\min_k \|c_i - c'_k\| \\ 0, & \text{otherwise.} \end{cases}$$

Let $F, \overline{F}$ be the original and the reconstructed node feature matrices, respectively. The node feature loss is defined as:

$$L_F = \frac{1}{n} \sum_{i=1}^n H(F_{i,:}, F'_{i,:}, w^N),$$

where $F' = \sigma \overline{F}$ is the reconstructed node feature matrix that is aligned with $F$; $w^N$ is the weighting vector for each coordinate; and $H(x, y, w)$ is the weighted cross entropy between probability vectors $x$ and $y$:

$$H(x, y, w) = -\sum_i w_i x_i \log(y_i).$$

Let $A, \overline{A}$ be the original and the reconstructed adjacency tensors, respectively. The edge loss is defined as:

$$L_E = \frac{1}{n(n-1)} \sum_{i=1}^n \sum_{j=1, j \neq i}^n H(A_{i,j,:}, A'_{i,j,:}, w^E),$$

where $A'$ is the reconstructed adjacency tensor that is aligned with $A$, satisfying $A'_{:,:,k} = \sigma \overline{A}_{:,:,k} \sigma^T$ for $k = 1, 2, ..., d_e$; and $w^E$ is the weighting vector for the edge classes. Note the exclusion of $j = i$ in the inner summation; self-loops are not counted.

We use the following weighting for the cross entropies:

$$w^N = \mathbf{1}, \quad w^E = \ln(\mathbf{1} \oslash D),$$

where $D$ is a probability vector that gives the empirical distribution of classes computed from data; and the operations $\oslash$ and $\ln$ are element-wise.

### D.3 NORMALIZNG FLOW LOSS $L_{NF}$

Let $z \in \mathbb{R}^D$ be a random vector with density $p_z(z)$ and let $x$ be a vector resulting from a sequence of $K$ invertible transformations from $z$:

$$x = f_K \circ f_{K-1} \circ ... \circ f_2 \circ f_1(z).$$

Define $z_0 = z$, $z_i = f_i(z_{i-1})$ for $i = 1, ..., K$, and $z_K = x$. By the change-of-variable formula, the density of $x$ can be expressed as

$$\log p_x(x) = \log p_z(z) - \sum_{i=1}^{K} \log \left| \det \left( \frac{\partial f_i}{\partial z_{i-1}} \right) \right|. \tag{1}$$

If $z$ is standard normal, then

$$\log p_z(z) = -\frac{1}{2}\|z\|^2 - \frac{D}{2} \log 2\pi. \tag{2}$$

In our work, we use RealNVP Dinh et al. (2016) to parameterize the mappings. Specifically, a mapping $f(u) = v$ is defined as

$$v_{1:d} = u_{1:d}$$
$$v_{d+1:D} = u_{d+1:D} \odot \exp(\text{MLP}_s(u_{1:d})) + \text{MLP}_t(u_{1:d}),$$

where $d = \frac{1}{2}D$. One easily calculates that the Jacobian of $f$ is

$$\frac{\partial f}{\partial u} = \begin{pmatrix} I & 0 \\ \frac{\partial v_{d+1:D}}{\partial u_{1:d}} & \text{diag}(\exp(\text{MLP}_s(u_{1:d}))) \end{pmatrix}.$$

Therefore,

$$\det \left( \frac{\partial f}{\partial u} \right) = \exp \left( \sum_{j=1}^{D-d} \text{MLP}_s(u_{1:d})_j \right). \tag{3}$$

Substituting equation 2 and equation 3 into equation 1, we obtain the negative log-lelilhood loss

$$-\log p_x(x) = \sum_{i=1}^{K} \sum_{j=1}^{D-d} \text{MLP}_s^i((z_{i-1})_{1:d})_j + \frac{1}{2}\|z_0\|^2 + \frac{D}{2} \log 2\pi. \tag{4}$$

### D.4 ADDITIONAL LOSS $L_R$

We follow the approach outlined in Ma et al. (2018) to introduce a soft valence regularization to penalize the excess of valence capacity for each atom, thereby encouraging the decoder to output chemically valid molecules. In particular, define $L_R = \sum_{i=1}^{n} \sigma(V(i) - U(i))$ for each molecule, where $\sigma$ is the ReLU function and $U(i)$ and $V(i)$ are the valence and capacity of atom $i$, respectively. The soft (thus differentiable) version is:

$$U(i) = \sum_{r} u(r) F'_{i,r},$$
$$V(i) = \sum_{j \neq i} \sum_{k} h(k) A'_{i,j,k},$$

where $F'$ and $A'$ are the reconstructed and aligned node feature matrix and adjacency tensor, respectively. Here, $u(r)$ is the valence of node type $r$, determined from data; and $h(k)$ is the capacity for edge type $k$. Specifically, $h(k) = 1, 2, 3, 1.5$ for single, double, triple, and aromatic bonds, respectively. Moreover, $u(r)$ and $h(k)$ are both 0 for phantom nodes and edges.

## E  $G^3$ PROPERTIES

A typical pitfall for many graph-based models is the ambiguity under node permutations. Reordering the nodes does not change the graph itself, but the latent representation of the graph may change if not properly modeled. Moreover, a single latent vector may not be able to decode an adjacency tensor with an arbitrary node permutation. Thanks to the geometry, our model $G^3$ is able to identify node correspondence and maintain desirable permutation and translation invariance.

**Proposition E.1.** *The G³ encoder and link predictor are invariant to input node permutations as well as translations on the coordinates.*

*Proof.* The G³ encoder is composed of a point encoder and a graph encoder. The point encoder is invariant to node permutations because the MLP is weight sharing for all points and the global pooling is permutation invariant. The graph encoder is also invariant to node permutations because the graph convolution layers of MoNet are. The overall latent vector is a concatenation of the outputs of these two encoders and thus is also invariant to node permutations. Additionally, the point clouds are preprocessed so that each is centered at the origin, invariant to translations.

The link predictor takes pairwise nodes as input and is therefore invariant to node permutations. In addition, node coordinates are used always in the form of displacements. Hence, the link prediction is always unchanged regardless of translation. □

## F  QUALITATIVE EXAMPLES

In Figure 5, we show the generated examples for QM9 as well as results of latent space interpolation. They correspond to the results reported in Table 1. The generated samples and interpolation results for ChEMBL250k are in Figure 6, which correspond to the results reported in Table 2.

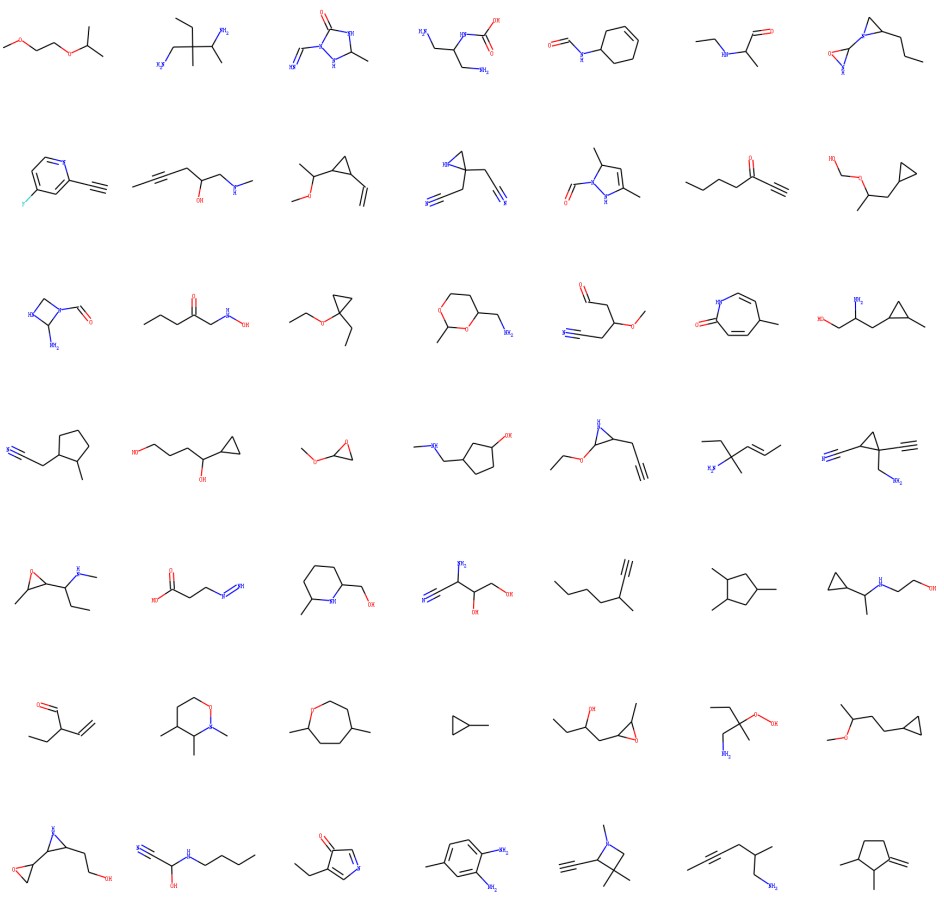

**Figure 5:** Generated molecules for QM9

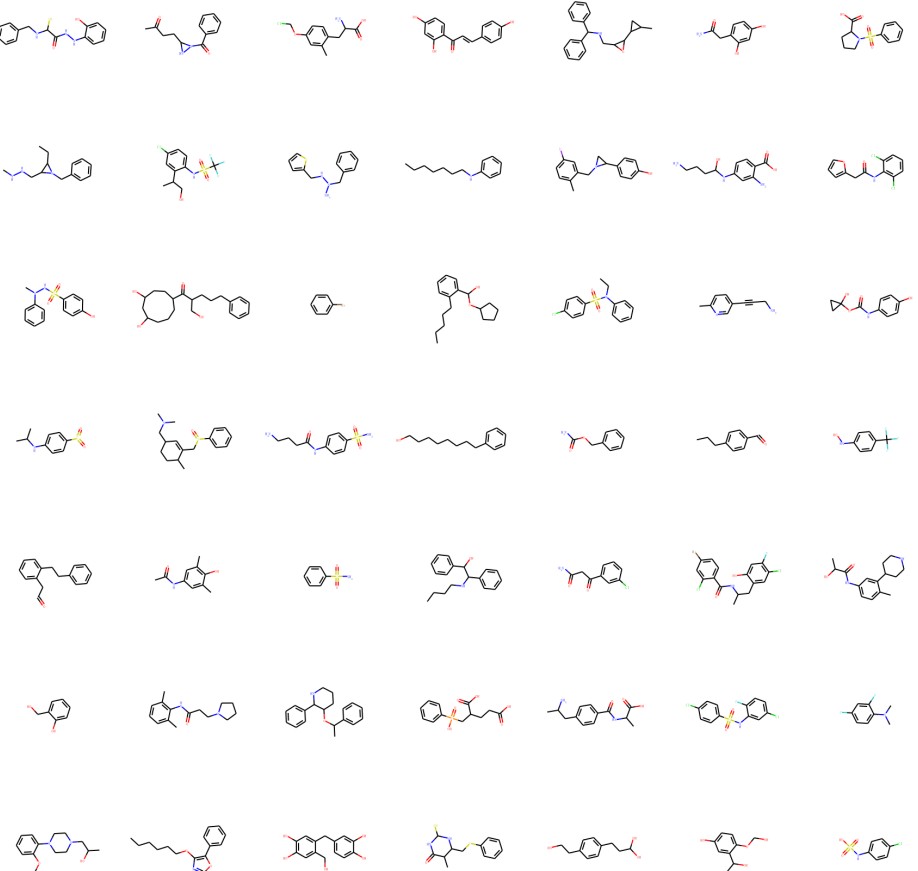

**Figure 6:** Generated molecules for ChEMBL250k

## G    EXPERIMENTS ON THE STANDALONE LINK PREDICTOR

In this section, we empirically demonstrate the effectiveness of our link predictor as a standalone model. The task is to predict the edge types for all edges in the graph given the ground truth node types and coordinates for all nodes. We train the link predictor on QM9 and ChEMBL250k until convergence, then report the accuracy and the F-1 score on the test set. The classification reports are computed by using scikit-learn, where the "ZERO" type corresponds to the non-existence of a bond. In addition, we provide visualized comparisons between the ground truth molecules and the reconstructed molecules with ground truth nodes and predicted edges.

| EDGE TYPE | PRECISION | RECALL | F-1 SCORE | SUPPORT |
|---|---|---|---|---|
| ZERO | 0.99999 | 0.99999 | 0.99999 | 778401 |
| SINGLE | 0.99855 | 0.99842 | 0.99848 | 198214 |
| DOUBLE | 0.98009 | 0.98150 | 0.98080 | 15246 |
| TRIPLE | 0.99972 | 0.99972 | 0.99972 | 7234 |
| AROMATIC | 0.99974 | 1.00000 | 0.99987 | 15592 |
| AGGREGATE METRICS | VALUE | VALUE | VALUE | SUPPORT |
| MACRO AVG | 0.99562 | 0.99593 | 0.99577 | 1014687 |
| WEIGHTED MACRO AVG | 0.99941 | 0.99940 | 0.99940 | 1014687 |
| ACCURACY | - | - | 0.99940 | 1014687 |

**Table 7:** Standalone G$^3$ link predictor's performance on the test set of QM9

| EDGE TYPE | PRECISION | RECALL | F-1 SCORE | SUPPORT |
|---|---|---|---|---|
| ZERO | 0.99984 | 0.99982 | 0.99984 | 37573897 |
| SINGLE | 0.97573 | 0.98228 | 0.97899 | 714020 |
| DOUBLE | 0.91250 | 0.87825 | 0.89505 | 89694 |
| TRIPLE | 0.98789 | 0.97759 | 0.98271 | 3838 |
| AROMATIC | 0.99904 | 0.99917 | 0.99910 | 633722 |
| AGGREGATE METRICS | VALUE | VALUE | VALUE | SUPPORT |
| MACRO AVG | 0.97500 | 0.96742 | 0.97114 | 39015171 |
| WEIGHTED MACRO AVG | 0.99920 | 0.99921 | 0.99920 | 39015171 |
| ACCURACY | - | - | 0.99921 | 39015171 |

**Table 8:** Standalone G$^3$ link predictor's performance on the test set of ChEMBL250k

We remark that while link prediction is pairwise on nodes, the inclusion of neighbor information allows it to faithfully reconstruct complex structures like aromatic rings. Overall, G$^3$'s link predictor attains near perfect aggregate F-1 score and accuracy on both QM9 and ChEMBL250k. It therefore serves as a viable standalone model for bond prediction in molecules.

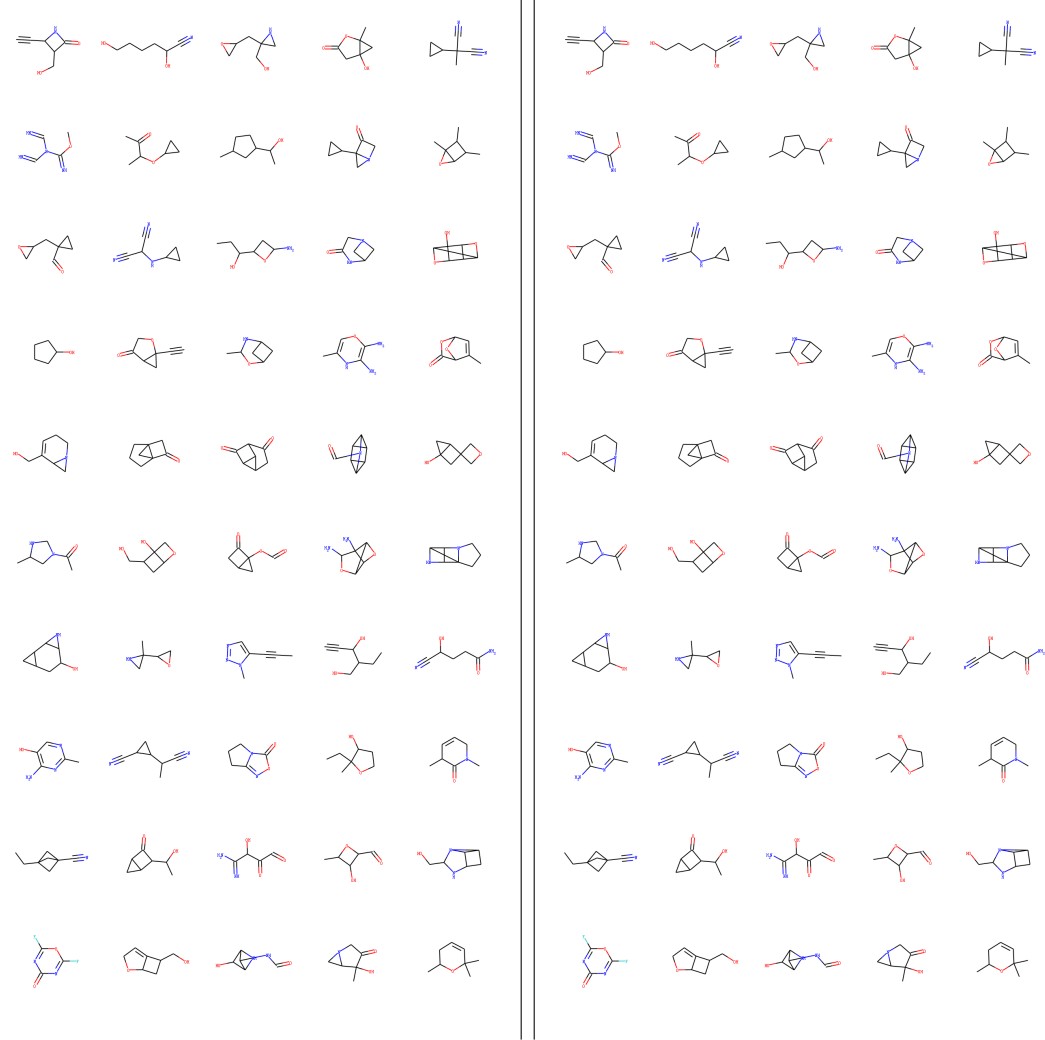

**Figure 7:** Left: 50 ground truth molecules from the QM9 test set. Right: the same molecules with edges reconstructed from the ground truth nodes by the link predictor. All 50 molecules in the left figure are reconstructed exactly.

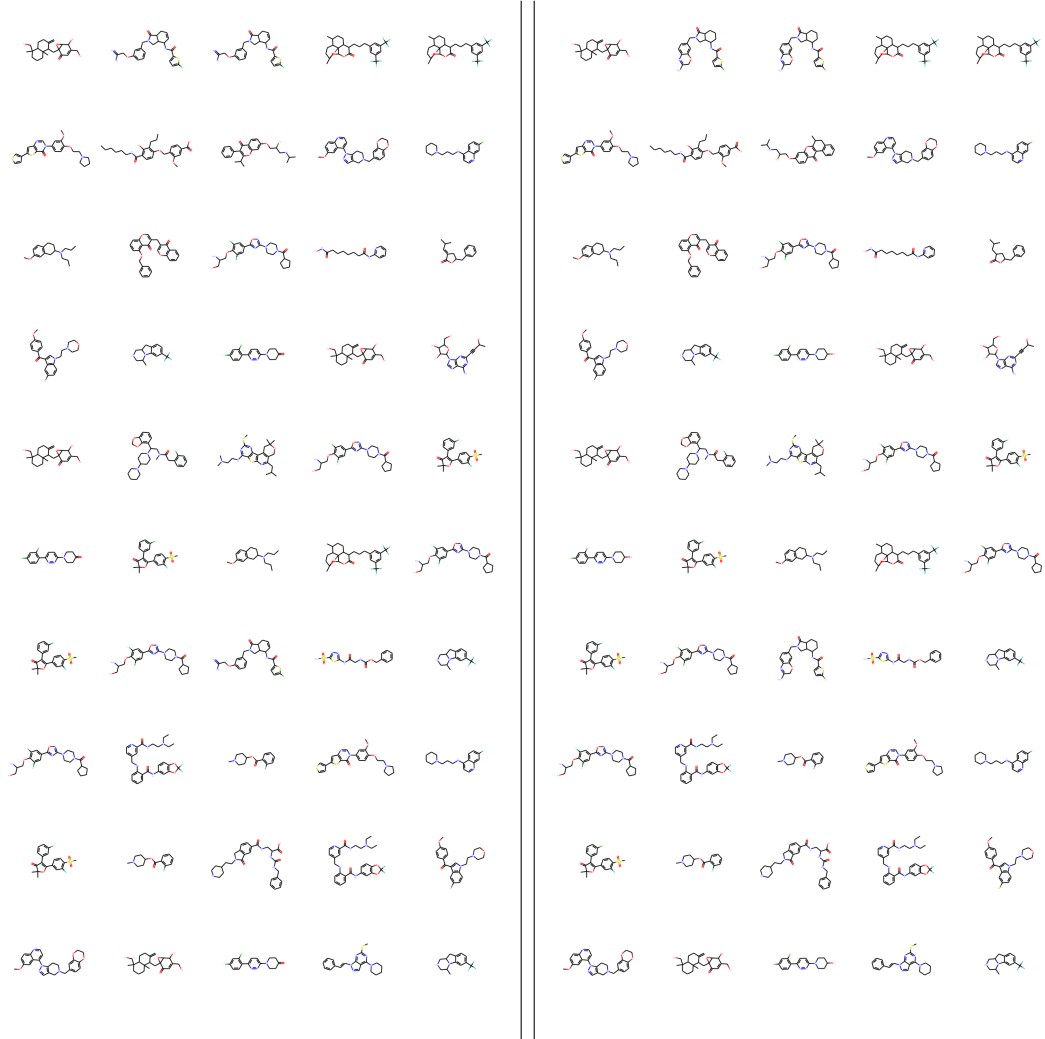

**Figure 8:** Left: 50 ground truth molecules from the ChEMBL250k test set. Right: the same molecules with edges reconstructed from the ground truth nodes by the link predictor. 46 out of 50 molecules in the left figure are reconstructed exactly.

# H  VISUALIZATION OF GEOMETRIC FEATURES

In this section, we provide qualitative examples for the learned geometric features in the point encoder. Similar to Wang et al. (2021), we present the channel-wise activation before the global pooling step in the point encoder of G$^3$ pretrained on ChEMBL250k. Intuitively, the difference in the level of activation reveals the geometric primitives that the neurons focus on.

In general, we observe that G$^3$'s point encoder can identify aromatic rings in molecules based purely on geometry; i.e. no bond information is provided. For example, Figure 9 shows that the 63rd neuron identifies rings to the right end of the molecules (see the red atoms).

In addition, different neurons can partition a molecule into meaningful sub-components. In Figure 10, we show neurons that are excited by the five-member ring on the left, the benzene ring in the middle, and the benzene ring on the right, respectively. In Figure 11, the top figure presents a neuron that identifies two rings on the right as a group, whereas the bottom figures showcase more discerning neurons that distinguish the two rings separately.

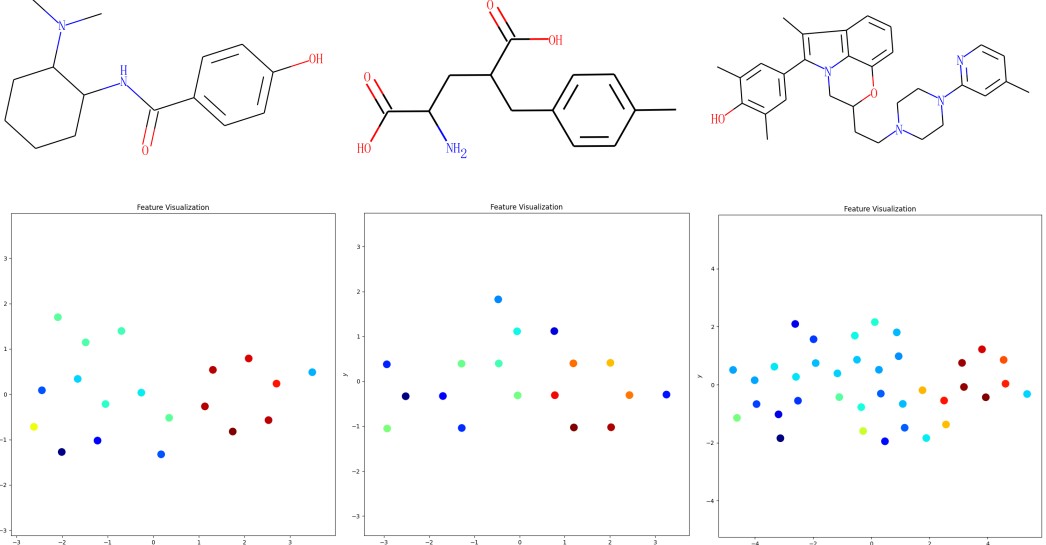

**Figure 9:** Consistency of neuron activation on six-member rings. Top row (from left to right): the molecular graphs of CN(C)C1CCCCC1NC(=O)c1ccc(O)cc1, Cc1ccc(CC(CC(N)C(=0)O)C(=O)O)cc1 and Cc1ccnc(N2CCN(CCC3Cn4c(-c5cc(C)c(O)c(C)c5)c(C)c5cccc(c54)O3)CC2)c1. Bottom row: the 63rd neuron's activation on the associate molecules, singling out (in red) the rings to the right.

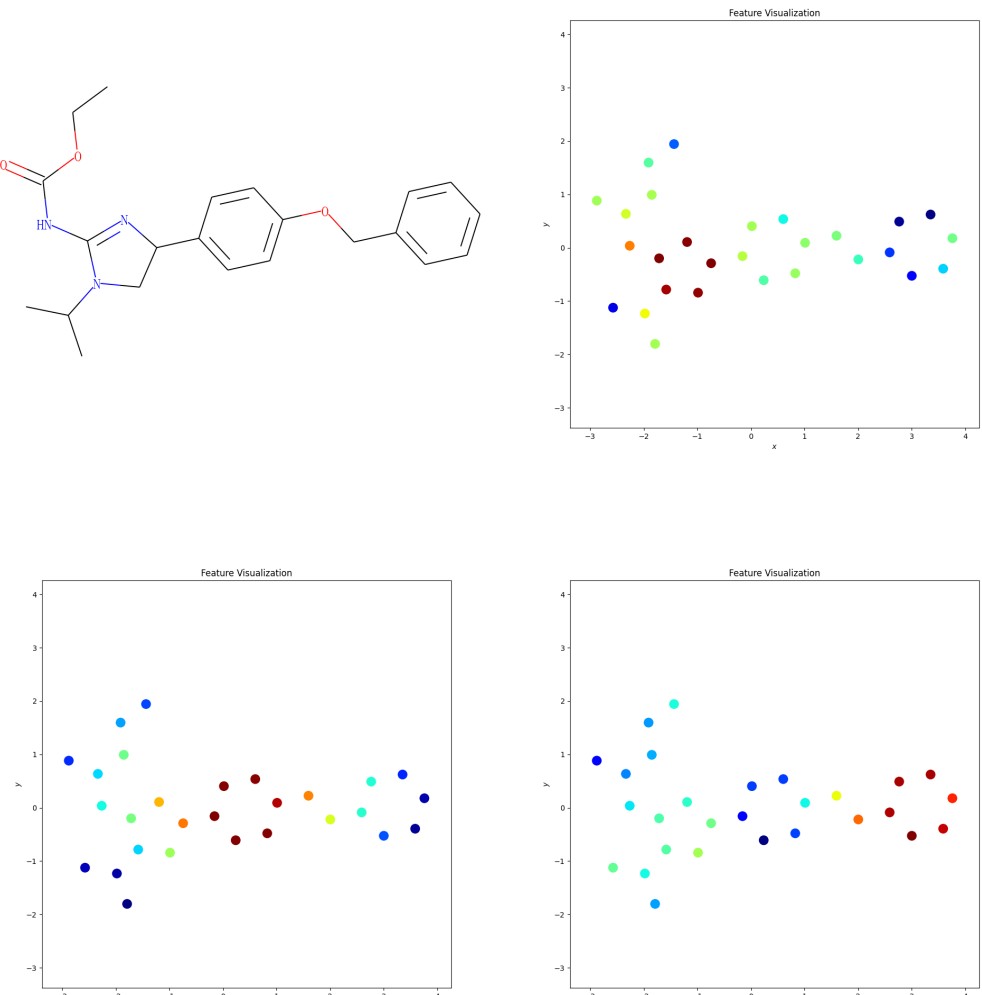

**Figure 10:** Top left: The molecular graph for CCOC(=O)NC1=NC(c2ccc(OCc3ccccc3)cc2)CN1C(C)C. Top right: the 172nd neuron's activation on this molecule as a point cloud, identifying the five-member ring on the left. Bottom left: the 241st neuron's activation on this molecules, identifying the benzene ring in the middle. Bottom right: the 571st neuron's activation on this molecule, singling out the benzene ring on the right. Red indicates higher activation values.

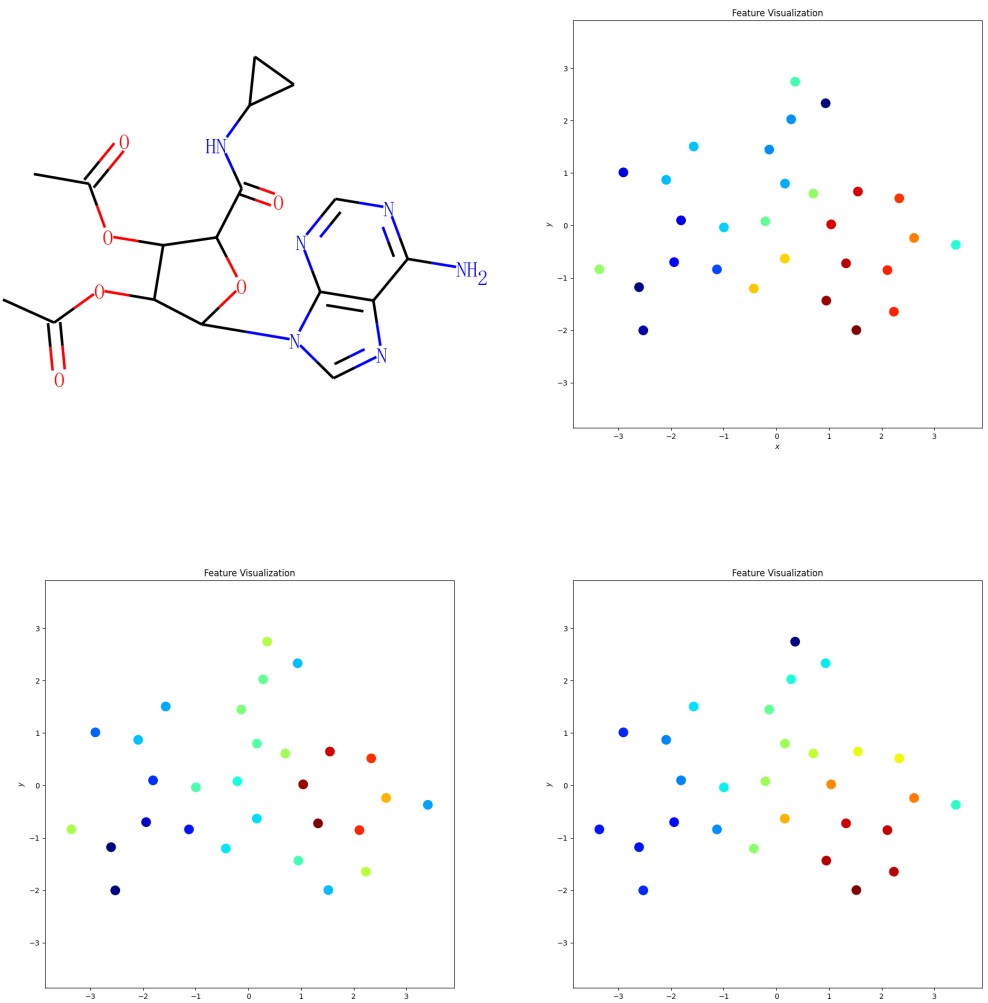

**Figure 11:** Top left: The molecular graph of CC(=O)OC1C(C(=O)NC2CC2)OC(n2cnc3c(N)ncnc32)C1OC(C)=O. Top right: the 56th neuron's activation on this molecule, identifying the two rings on the right as a group. Bottom left: the 663rd neuron's activation on this molecule, singling out the six-member ring on the right. Bottom right: the 670th neuron's activation on this molecule, identifying the five-member ring on the right. Red indicates higher activation values.

