# OpenReview forum: "$G^3$: Representation Learning and Generation for Geometric Graphs"
_ICLR.cc/2022/Conference — ICLR 2022 Submitted_

### Official Review · Reviewer_PjEF · 2021-11-02

**Correctness:** 4
**Technical Novelty And Significance:** 3
**Empirical Novelty And Significance:** 3
**Recommendation:** 8
**Confidence:** 4

**Main Review:**

**Strength**

1. It is true that restoring a graph is challenging. Works [1][2] at the same period tackle it via matching the representations in the latent space instead of defining reconstruction loss on the data space. This paper proposes a different way via separating (or..cascading) 3d and feature decoding processes, whose effectiveness and efficiency are empirically validated.

2. a good integration of previous works, especially connecting generative models in point clouds and molecules.


**Weakness**

1. Link predictor is quite interesting (and overselling), more empirical results and interpretations will be helpful to support the claim of "We highlight that our link predictor works remarkably well as a standalone module for molecules, ..."
2. Unclear when interpreting tables 1 & 2. How to determine the angle directions? In comparisons with all the baselines or a certain subset of baselines (e.g., methods without using hard valency checks)?
3. Table 3 seems not to well support the claim of "It is clear that both ablated models are inferior to G 3 in nearly all molecular metrics, which attests to the efﬁcacy of G 3 ’s overall architecture.", for instance, Validity and Novelty are decreasing when combining both encoders, and so on. More elaborations?

**Comments**

1. What do you mean by the term "large inner product in 3D" (page 5)?
2. It would be helpful to bold the best results in the tables.

**Open Questions**

Not addressing the following comments won't have negative effects on my rating.
But it would be nice if the authors can make a few clarifications, which also helps with my understandings!
1. explorations on the template design in the coordinate decoder
2. interpretability on the learned geometrical information in the point cloud encoders for molecules (something like [3])
3. a better (analytical/empirical) way to support the claim that geometry and structure are "complementary"? I don't find an inspiring explanation in [1] or in this draft.

**Reference**

1. Pre-training Molecular Graph Representation with 3D Geometry, arXiv 2021
2. 3D Infomax improves GNNs for Molecular Property Prediction, arXiv 2021
3. Unsupervised Point Cloud Pre-Training via Occlusion Completion, ICCV 2021

**Summary Of The Paper:**

This paper is well written and focuses on an interesting problem, to design a better framework for a molecular generation while incorporating 3D geometry, using connections between 3D point clouds and GNNs.

**Summary Of The Review:**

In general, I like the idea but given the fact a lot of concerns need to be addressed, I tend to give a borderline.

However, I am willing to adjust my scores based on the reviewer's reply.

---

> ### Author Response · Authors · 2021-11-20
> **Clarifications for our claims and additional experiments for the link predictor.**
>
> Thank you for your insightful comments and questions, we answer them below in the order that they're raised.
>
> &nbsp;
>
> > Q1: Link predictor is quite interesting (and overselling), more empirical results and interpretations will be helpful to support the claim of "We highlight that our link predictor works remarkably well as a standalone module for molecules, ..."
>
> We have included more numerical experiments on the standalone predictor in section G of the updated paper. We believe the results convincingly justify the effectiveness of the link predictor on its own. Additionally, for comparison under context, we note a well-known alternative for bond prediction---Open Babel. Prior work such as G-SchNet uses this software to reconstruct bonds based on atomic coordinates. As noted by the authors (see, e.g., https://github.com/atomistic-machine-learning/G-SchNet), Open Babel is not sufficiently reliable for bond reconstruction, especially for aromatic rings containing carbon and nitrogen atoms. To ameliorate the lack of robustness, the authors employed handcrafted heuristics as supplements to Open Babel and obtained satisfactory results on the QM9 dataset. Nonetheless, they noted that the heuristics might not generalize well to other datasets. In contrast, our link predictor can reliably predict bonds for both datasets we experimented with.
>
> &nbsp;
>
> > Q2:  Unclear when interpreting tables 1, 2. How to determine the angle directions? In comparisons with all the baselines or a certain subset of baselines (e.g., methods without using hard valency checks)?
>
>
> To clarify, the arrows indicate the favorable direction for each metric: upward arrows mean the higher the better, while downward arrows mean the lower the better. We apologize for the confusion. The arrows do not suggest whether our method is better than the baselines or worse.
>
> &nbsp;
>
> > Q3: Table 3 seems not to well support the claim of "It is clear that both ablated models are inferior to G3 in nearly all molecular metrics, which attests to the efficacy of G3 ’s overall architecture.", for instance, Validity and Novelty are decreasing when combining both encoders, and so on. More elaborations?
>
>
> Our conclusion of the superiority of the combined encoder is based on a majority of the metrics, but not all. Note that some metrics are at odds with others. For example, one can easily overfit the training set and saturate validity and novelty, by sacrificing uniqueness (for example, repeatedly producing the same valid and novel molecule, leading to poor uniqueness). Hence, a holistic view of all metrics properly measures the performance.
>
>
> &nbsp;
>
> > Q4: What do you mean by the term "large inner product in 3D" (page 5)?
>
> Traditional link predictors use the inner product of two feature vectors to determine if an edge exists between the two corresponding nodes. However, for geometric graphs, such an inner product approach is less meaningful, when the node features contain 3D coordinates. One way to see this is that the link predictor is expected to predict the same edge regardless translation. However, inner products are not translation invariant.
>
> &nbsp;
>
> > Q5: explorations on the template design in the coordinate decoder
>
> We view the template design a form of domain-specific inductive bias. Conceptually speaking, the goal is to use a template such that it can be transformed to a given point cloud ``with ease''; i.e., the parameterized transformation is easy to learn. We have experimented with three templates: a 2D grid, a (quasi-)uniform sampling on the 3D unit sphere, and a (quasi-)uniform sampling on the 2D unit disc. We find that the unit sphere works best for QM9, while the unit disc works best for ChEMBL. This is probably because the coordinates offered by the datasets are 3D in QM9 and 2D in ChEMBL.

---

> > ### Comment · Reviewer_PjEF · 2021-12-06
> > **Further response**
> >
> > Thanks for the updates and clarifications, especially on the link predictor (Appendix G) and interpretability (Appendix H).
> > Unlike reviewer qSkw, I think the novelty is enough while the techniques and the experiments are grounded.
> > I've raised my original recommendation for an apparent acceptance.

---

> ### Author Response · Authors · 2021-11-20
> **Visualization of geometric features**
>
> > Q6: A better (analytical/empirical) way to support the claim that geometry and structure are "complementary"? I don't find an inspiring explanation in [1] or in this draft.
>
> A focus of this work is the dual processing of the geometric and the structural information. Ignoring either piece leads to suboptimal representations and generation. On the encoder side, ablation studies suggest that a combined encoder works better than a single encoder (Table 3). On the decoder side, our innovative link predictor which reconstructs structure through first reconstructing the geometry also demonstrates superior performance (Section G). Furthermore, many of the baselines we compare with deal with either geometry (e.g., G-SchNet) or structure (e.g., GraphAF and GraphNVP), but not both. G3's superior performance over these baselines also corroborates that a single side of the information is insufficient to capture to nature of the object.
>
> &nbsp;
>
> > Q7: Interpretability on the learned geometrical information in the point cloud encoders for molecules (something like [3])
>
> We conducted an additional experiment, using the same procedures as the reference you provided, and observed similarly interpretable patterns. Specifically, the extracted features from the point encoder distinguish aromatic rings from other parts of a molecule. Details have been included in Section H of the updated paper.
>
> &nbsp;
>
> Thank you for the time you have taken to review our work. We hope the above response provides more evidence of the effectiveness of our link predictor and clarifies claims.

---

### Official Review · Reviewer_rFAq · 2021-11-02

**Correctness:** 4
**Technical Novelty And Significance:** 2
**Empirical Novelty And Significance:** 3
**Recommendation:** 5
**Confidence:** 2

**Main Review:**

Strengths: The run time of G3 is impressive compared with strong baselines such as  GraphAF and JT-VAE due to the one-shot nature while obtaining as good in quality as these baseline methods. The designed method may be applied to the application of drug discovery, etc.

Weaknesses: The technical contribution is unclear/limited. The comparison (from both the theoretical side and architecture design side) between G3 and other popular baseline methods is missing. It is unclear why G3 is better than others and why it is so efficient but not other methods. How this designed architecture is useful to guide other works? The other concern is the generalization ability of the proposed method since the proposed method was only tested on two datasets.

**Summary Of The Paper:**

This paper studies the problem of geometric graph generation (mainly focusing on molecule graph generation). Specifically, the authors propose a new method,  namely Geometric Graph Generator (G3), which generates three-dimensional geometric graphs. Different from others, G3 can capture both the combinatorial and the geometric information of graphs. The proposed algorithm can successfully generate molecule graphs and runs faster than other baseline methods. The key parts of G3 are the point encoder and graph encoder, and the three-step decoding process, that is the coordinate decode, feature decoder, and finally graph decoder.  The experimental results show that G3 has better performance than other generators in terms of validity, uniqueness, and novelty.


**Summary Of The Review:**

Overall, the problem studied is interesting. The authors aim at designing new NN architecture to generate geometric graphs. The experimental results look promising. Moreover, G3 is much faster than current strong baseline methods.

However, the generalization ability of the proposed architecture to other datasets is unknown. The encoding part seems heavily based on other works (similarly decoder parts). Compared with other methods, the reviewer does not see the real novelty part of the technical section. (The problem of graph generation is out of my research area. I may miss something important.)

---

> ### Author Response · Authors · 2021-11-20
> **Clarifications on the technical contributions of G3**
>
> Thank you for your insightful comments and questions, we answer them below in the order that they're raised.
>
> &nbsp;
>
> > Q1: The technical contribution is unclear/limited. The comparison (from both the theoretical side and architecture design side) between G3 and other popular baseline methods is missing. The encoding part seems heavily based on other works (similarly decoder parts). Compared with other methods, the reviewer does not see the real novelty part of the technical section. (The problem of graph generation is out of my research area. I may miss something important.)
>
> We believe that the main novelty of this work is the artful combination of geometry and structure. Prior work on molecular generation either considers the geometry alone (e.g., G-SchNet) or the graph structure alone (e.g., GraphAF and GraphNVP), but not both. G3 is an architecture that organically combines both sides of the information. As a result, G3 generates molecules with better quality than do methods that process only a single side of the information.
>
> Furthermore, the philosophy of combining graph structure and geometry  leads to a novel decoding procedure for a graph. In general, graph decoding is challenging; traditional methods either decode nodes and edges sequentially or decode the graph adjacency matrix in one shot. The former approach suffers scalability because learning a long sequence is hard, while the latter approach suffers lack of permutation equivariance because nodes are implicitly ordered in an adjacency matrix. In contrast, G3 decodes a graph through reconstructing the geometry first, which is a novel procedure rarely seen in the literature.
>
> &nbsp;
>
> > Q2: It is unclear why G3 is so efficient but not other methods.
>
> G3's speed advantage is the most pronounced when compared to sequential models like GraphAF and JT-VAE. These models decode graphs incrementally (node-by-node or motif-by-motif) and are hard to parallelize by GPU. Furthermore, they employ validity checks during generation, which incur additional overhead. For example, GraphAF samples a new edge from the learned distribution at each sequential step and performs one valcency check. In case of invalidity, backtracking is applied. These steps are highly sequential and are a major bottleneck to parallelization. In contrast, G3 decodes the graph in one shot (through link prediction), which is rather efficient and GPU friendly. Other one-shot generation models, e.g., GraphNVP, are similar to G3 in computational efficiency, but they produce inferior results as shown in Tables 1 and 2.
>
> &nbsp;
>
>
> > Q3: How this designed architecture is useful to guide other works?
>
> The take-home message of this paper is to incorporate both geometry and structure when performing representation learning and learning generative models. Such a design philosophy has demonstrated superior performance over the negligence of geometry or structure (see comparisons with baseline models as well as ablation studies). Many components of the proposed architecture borrow existing neural networks, but the organic composition of them driven by the philosophy sets up a reference to compete with for future works on geometric graphs.
>
> &nbsp;
>
> > Q4: The other concern is the generalization ability of the proposed method since the proposed method was only tested on two datasets.
>
> The two datasets we use, QM9 and ChEMBL, are representative in the literature of graph generative models. ChEMBL is one of the largest benchmarks, in terms of graph size and the number of graphs, for testing scalability. Many models working well for QM9 face challenges extending their good performance to ChEMBL. By the standard of the literature, many of the representative methods, including the baselines we compare with, are validated on at most two molecule datasets, typically QM9 and ChEMBL/ZINC. For example, SDVAE is evaluated on ZINC; MolGAN is evaluted on QM9; GraphNVP is evaluated on QM9 and ZINC; and JT-VAE is evaluted on ZINC. ZINC's scale is very similar to our portion of ChEMBL, but since it does not offer geometry information, we omit this dataset.
>
> &nbsp;
>
> Thank you for spending the time and effort to review our manuscript. We hope that our responses shed more light into the most important contributions of this work.

---

### Official Review · Reviewer_qSkw · 2021-11-03

**Correctness:** 3
**Technical Novelty And Significance:** 2
**Empirical Novelty And Significance:** 2
**Recommendation:** 3
**Confidence:** 3

**Main Review:**

The paper is clearly written and well motivated, and easy to follow. The network is comprised of multiple parts which are clearly defined and explained.
The numerical experiments indeed show an improvement over other methods, however not in all metrics.

I have a few concerns:
1. If I understand correctly, the authors choose to first learn the geometry, via template folding (as in FoldingNet), and then, once the node features are learned, use a link prediction mechanism to predict the topology of the graph. To the best of my knowledge, the approach of template folding is bound to its original topology (e.g. genus 0), which is limited. It would be beneficial and enriching if the authors can elaborate on this point. Were other methods considered to learning the geometry, and perhaps the authors can state what is the influence of using such template, which in many cases may not fit to the actual topology of the underlying graph.

2.How did the authors choose the link prediction mechanism? In what sense is it different/better than using something like GAT and its variants ?

3.The authors present their results on 2 popular datasets. However, in order to reach to a more conclusive evidence of the importance and contribution of this work, I would suggest the authors to add more applications, such as learning the geometry and topology of meshes (e.g., ModelNet, ShapeNet). Also, the task of protein folding (e.g., AlphaFold) seems to be appropriate for the suggested method.



**Summary Of The Paper:**

The authors propose to combine point cloud and graph encoders with coordinate and feature decoders in order to learn the geometry first, and then try to predict the adjacency matrix of the graph, to obtain a better overall graph representation of the considered applications.


**Summary Of The Review:**

The authors propose an interesting combination of known methods to learn the geometry and topology of the  graph for molecular-based applications. The results reveal that some improvements over existing method is obtained. Some of the limitation could be further discussed, and more experiments would strengthen the contribution of the paper.

---

> ### Author Response · Authors · 2021-11-20
> **Clarification on G3's methodology**
>
> Thank you for the astute observations and questions, we answer them below in the order that they're raised.
>
> &nbsp;
>
> > Q1: If I understand correctly, the authors choose to first learn the geometry, via template folding (as in FoldingNet), and then, once the node features are learned, use a link prediction mechanism to predict the topology of the graph. To the best of my knowledge, the approach of template folding is bound to its original topology (e.g. genus 0), which is limited. It would be beneficial and enriching if the authors can elaborate on this point. Were other methods considered to learning the geometry, and perhaps the authors can state what is the influence of using such template, which in many cases may not fit to the actual topology of the underlying graph.
>
> Indeed, our decoder works as you described. However, we would like to note a subtle difference between the topology of a graph and the topology of a surface (represented by a point cloud). For surfaces, genus is a topological definition that characterizes surface types; while for graphs, topology is often a synonym of ``structure''---it refers to connections between node pairs (i.e., edges). Template folding is a tool we use to transform, in a parameterized manner, a starting configuration of points to various terminal configurations; but the graph structure is recovered by the link predictor instead. The atoms might not be feasibly interpreted as points on a surface. That said, this parameterized tool fits intuition because it does not only works for a genius 0 surface. For example, a followup work of FoldingNet, the TearingNet, adapts the template folding idea for more complex configurations, including point clusters (see, e.g., the bottom example of Figure 1 of https://arxiv.org/pdf/2006.10187.pdf). Our case, atoms, is more like clusters, if we consider each atom to be one. Then, the intuition is that proper tearing of the template is able to learn well the separation of atoms.
>
> &nbsp;
>
> > Q2: How did the authors choose the link prediction mechanism? In what sense is it different/better than using something like GAT and its variants ?
>
> Our link prediction mechanism originates from one of the most common approaches to predicting existence of edges: starting from representations of a pair of nodes, take the inner product and apply a sigmoid. Of course, the simple inner product form is infeasible for our case, because the node representation includes atomic coordinates but inner products do not preserve translation invariance. Hence, we design the link predictor to satisfy various desirable properties: being translation invariant, being neighborhood aware, and being symmetric. These properties apply to general geometric graphs. Additionally, for molecules, we also inject chemistry into the design: a bond depends additionally on the types of atoms, their distance, and their aromatic ring membership.
>
> There are other ways to design the link predictor, as you suggested. GAT and its variants use the attention mechanism to determine link probability. However, the vanilla GAT acts on a known graph and the attention merely computes different weightings on the known edges. A straightforward variant would ignore the given graph and compute all-pair attention. However, such an approach is equivalent to running GAT on the complete graph, wherein links are determined globally by information of the entire molecule. This is in striking contrast to chemistry: bonds are local and they do not depend on irrelevant atoms far away.

---

> ### Author Response · Authors · 2021-11-20
> **Mesh datasets and protein folding are out of the scope of G3**
>
> > Q3: The authors present their results on 2 popular datasets. However, in order to reach to a more conclusive evidence of the importance and contribution of this work, I would suggest the authors to add more applications, such as learning the geometry and topology of meshes (e.g., ModelNet, ShapeNet). Also, the task of protein folding (e.g., AlphaFold) seems to be appropriate for the suggested method.
>
> While we agree that meshes and protein folding are important applications of geometric graphs, the problem of molecular representation and generation we focus on in this work is already quite complex, for which we have conducted extensive validations on the effectiveness of the proposed framework (measuring distributions, predicting and optimizing properties, investigating latent spaces, and comparing ablated architectures). Each other application has their unique complications that by themselves justify separate investigations beyond the scope of this paper. For example, meshes have not been shown to be as diverse as molecules in the sense that interpolation of the latent space results in meaningful, novel objects; and the problem of protein folding does not involve the generation of new proteins. Rather, this work sets the philosophy of leveraging both geometry and structure information whenever available and sets a good reference point for the extension to other applications.
>
> &nbsp;
>
> Thank you for taking the time to review our work. We hope our responses have clarified the link predictor design as well as resolving your other concerns.

---

### Official Review · Reviewer_LcmV · 2021-11-04

**Correctness:** 4
**Technical Novelty And Significance:** 3
**Empirical Novelty And Significance:** 3
**Recommendation:** 8
**Confidence:** 4

**Main Review:**

Overall this is a well-written paper with clear motivations and description of each model component. The problem of generating geometric graphs is important and the paper demonstrates that the proposed G^3 can leverage both graph structures and the point cloud geometry well. The experiments are extensive and the results are clear with domain-knowledge interpretation.

On the negative side, it looks like many components of G^3 are be studied before, hence reducing the novelty of the paper.

**Summary Of The Paper:**

The paper proposes a generative models for learning over the space of geometric graphs -- those graphs whose nodes are associated with geometric coordinates (point clouds). The point cloud serves as basis for decoding graph structure. This makes the entire system efficient. An extensive suite of experiments on chemical benchmarks (QM9 and ChEBML) show that the proposed method is competitive against state-of-the-art rivals on a variety of measures and tasks.


**Summary Of The Review:**

This is a nice paper on an important topic and with a successful proposal evaluated extensively. The novelty may be less than ideal due to the integration of existing ideas.

---

> ### Author Response · Authors · 2021-11-20
> **Clarifications on G3's novelty**
>
> Thank you for the kind comments, we would like to provide some thoughts on the technical novelty of G3.
>
> &nbsp;
>
> > Q1: On the negative side, it looks like many components of G3 are be studied before, hence reducing the novelty of the paper.
>
> While several modules in G3 are inspired by existing neural networks, our main contribution is the artful combination of geometry and structure information for representation learning and generation. In particular, prior works on molecular generation consider the molecule either as a graph only (e.g., GraphAF) or as a point cloud only (e.g., G-SchNet). G3 meaningfully leverages both sources of the information. In addition, such processing leads to a novel decoding procedure that recovers the structure through first recovering the geometry. Such a decoder has unique advantages. In general, graph decoding is a rather challenging task because of its combinatorial nature. Traditional approaches either decode nodes and edges sequentially or decode the graph adjacency matrix in one shot, both of which pose significant drawbacks (for sequence learning it does not scale well with respect to the sequence length, while for adjacency matrices permutation equivariance is lost). Our novel decoding procedure contributes to the effectiveness and the efficiency of the entire framework.
>
> &nbsp;
>
> Thank you for taking the time and effort to review our manuscript. We hope the above discussion provides additional insights into the contributions of our work.

---

> > ### Comment · Reviewer_LcmV · 2021-11-30
> > **Response**
> >
> > Thanks for clarification on the novelty part. I will keep my score.

---

### Decision · Program_Chairs · 2022-01-20

**Decision:**

Reject

**Comment:**

This paper presents a generative model for geometric graphs.  The main contribution is to separate the representation and generation of geometry from that of graph structure and features.  Based on this idea the authors assembled a set of existing ideas and built an auto-encoder style generative model for geometric graphs.

This paper sits on the borderline, with reviewers split on both sides.  I appreciate the clarifications from the authors during the rebuttal and the interactions with the reviewers.  The main concern is the novelty of this approach, as the main contribution is the idea of separating geometry from graph structure, and most other components of the pipeline already exist in the literature.  Because of this I think the paper can probably devote a bit more to this ablation study.  In particular the paper currently lacks detail about whether the size of the models were controlled when doing the ablation, which could be a confounding factor that explains why the joint model with both geometry and graph structure works better.  Also the different architecture choices may also factor into the difference, it would be more convincing if for example the same combination of multi-head attention blocks and GINE networks are used for the ablated graph encoder (you can simply concatenate the features from both on all layers, or even at the end).

Based on this I would recommend rejection at this time but encourage the authors to improve the paper and send it to the next venue.